

# neo4jsbml: import systems biology markup language data into the graph database Neo4j

Guillaume Gricourt[1], Thomas Duigou[1], Sandra Dérozier[2] and Jean-Loup Faulon[1]

[1] Université Paris-Saclay, INRAE, AgroParisTech, Micalis Institute, Jouy-en-Josas, France
[2] Université Paris-Saclay, INRAE, MaIAGE, Jouy-en-Josas, France

## ABSTRACT

Systems Biology Markup Language (SBML) has emerged as a standard for representing biological models, facilitating model sharing and interoperability. It stores many types of data and complex relationships, complicating data management and analysis. Traditional database management systems struggle to effectively capture these complex networks of interactions within biological systems. Graph-oriented databases perform well in managing interactions between different entities. We present neo4jsbml, a new solution that bridges the gap between the Systems Biology Markup Language data and the Neo4j database, for storing, querying and analyzing data. The Systems Biology Markup Language organizes biological entities in a hierarchical structure, reflecting their interdependencies. The inherent graphical structure represents these hierarchical relationships, offering a natural and efficient means of navigating and exploring the model's components. Neo4j is an excellent solution for handling this type of data. By representing entities as nodes and their relationships as edges, Cypher, Neo4j's query language, efficiently traverses this type of graph representing complex biological networks. We have developed neo4jsbml, a Python library for importing Systems Biology Markup Language data into a Neo4j database using a user-defined schema. By leveraging Neo4j's graphical database technology, exploration of complex biological networks becomes intuitive and information retrieval efficient. Neo4jsbml is a tool designed to import Systems Biology Markup Language data into a Neo4j database. Only the desired data is loaded into the Neo4j database. neo4jsbml is user-friendly and can become a useful new companion for visualizing and analyzing metabolic models through the Neo4j graphical database. neo4jsbml is open source software and available at https://github.com/brsynth/neo4jsbml.

## INTRODUCTION

Genome-Scale Metabolic Models (GEMs) of biological systems are commonly represented using the Systems Biology Markup Language (SBML) format. This format is actively maintained and updated, especially, by adding new features to meet new needs. Using a standard format facilitates interoperability repeatability, and reproducibility. Several standards have emerged in systems biology (*Shin et al., 2023*). The CellML standard stores computer-based mathematical models (*Clerx et al., 2020*) and the Simulation

Corresponding author
Guillaume Gricourt,
guillaume.gricourt@inrae.fr

Experiment Description Markup Language (SED-ML) (*Waltemath et al., 2011*) enables the reproduction of simulation experiments. The use of a standard format to represent data enables the creation of a rich software environment such as COPASI (*Hoops et al., 2006*), Tellurium (*Choi et al., 2018*) and MASSpy (*Haiman et al., 2021*) for the construction, simulation, and visualization of dynamic metabolic models. A wide range of issues can be addressed in this way, including the construction of metabolic pathways (*Shen et al., 2020*), study of secondary metabolism in bacteria (*Qiu, Yang & Zeng, 2023*) and representation of cell behavior and interactions in cancer (*Kazerouni et al., 2020*). To represent these types of biological processes, the SBML format encodes several nested components and their interactions in an XML-based document. The standard represents complex systems by organizing the data as components. To illustrate, biological systems can contain thousands of metabolites and reactions, leading to a high level of complexity in their interactions (*Hucka et al., 2019*). The SBML Level 3 standard is well-suited for describing reaction-based models. To represent different types of models, such as constraint-based, logical network, and rule-based models, the SBML standard has been enhanced using different packages. As described in their specifications, these packages add components or properties to the model.

Neo4j is a powerful graph database that enables the storage, querying, and analysis of large amounts of data. Unlike relational databases, Neo4j leverages the power of graph theory, representing data as nodes or relationships which embed properties to store additional information. The graph-based structure is well-suited to tackle intricate relationships and decipher complex problems, such as social networks and network analysis. The graph query language, named Cypher, allows querying the graph to retrieve connected data and perform create, read, update, or delete operations on the database. Neo4j ensures data integrity and consistency by respecting the ACID principle (*Meier & Kaufmann, 2019*). Specific capabilities can be added to the Neo4j database through a software extension, a plugin, enabling users to customize and enhance its features for dedicated applications. The Neo4j database comes with a broad ecosystem which includes tools like Arrows for designing the database structure, numerous drivers for communicating with the database, building queries intuitively with Cypher and Bloom for visualizing data.

With Neo4j, metabolic models can be represented as a network of nodes and relationships, where each node represents a biological or supporting object within a component, such as a metabolite, reaction, or unit definition. Each relationship serves as a connection between two metabolic reactions, enabling easy exploration of different pathways and interactions between human metabolic data (*Balaur et al., 2016*). Using Neo4j with metabolic models can provide valuable insights into complex interactions, as it allows for the integration of heterogeneous data, such as chemical species, reactions, enzymes, and taxonomic data. This integration can aid in identifying key players across a wide range of biological application domains (*Swainston et al., 2017*).

More recently, based on the graph database MaSyMoS, which stores SBML and CellML models to represent biological systems in terms of functional, behavioral, and structural aspects (*Henkel, Wolkenhauer & Waltemath, 2015*), a protocol was developed to compare biochemical reaction networks (*Lambusch et al., 2018*). Nonetheless, analyzing its own

SBML data using Neo4j remains a challenge. Biochem4j provides a web interface that does not allow users to analyze their data in their database. Recon2Neo4j and MaSyMoS created Java software using the JSBML library (*Rodriguez et al., 2015*) to import the data into their database. However, data loading is performed according to a predefined schema that retain several SBML components or properties that are targeted by their applications.

We are presenting neo4jsbml, a user-friendly Python package to import SBML files into a Neo4j database. Neo4jsbml utilizes tools developed by Neo4j to define the database schema, load data, query entities and relationships, and visualize the created graph. This package offers flexibility, extensibility, and ease of use in combining graph databases with SBML files. Flexibility is provided by leaving the choice of entities to be analyzed by the researcher. Extensibility is based on the intrinsic operation of neo4jsbml *via* introspection. Finally, neo4jsbml can be used through the command line and requires no programming skill. The graph can be enriched with heterogeneous data, allowing researchers to build, manipulate, annotate, and store their data with greater efficiency.

## MATERIAL AND METHODS

### Implementation

SBML data are structured according to the specifications defined by *Hucka et al. (2019)*. This file format has a skeleton of main components, but optional data can be added through the use of packages. The SBML specification Level 3 Version 2 describes eleven components: function definitions, unit definitions, compartments, species, parameters, initial assignments, rules, constraints, reactions, events and the model. Each component has a specific role and stores specific and generic information. The model component is special, it serves as a container for the other components but also supports some information, such as the identifier of the model, which can be exploited.

Sometimes, not all components are suitable for analysis. To retain the desirable data from SBML documents, neo4jsbml utilizes the concept of introspection. Introspection allows a program to examine an object's characteristics, such as its name, properties, and methods, making the program more efficient and robust. It is a powerful feature of object-oriented languages that exposes details about objects at runtime, and Python ships with a few built-in functions for this purpose. When importing SBML data into Neo4j, introspection is used to automate the extraction of properties and associated structured values of items mentioned in a definition schema. In this way, the user filters the main components and attributes to be loaded into the database.

### Running method

The usage of the tool is described in Fig. 1.

It takes place in two stages. First of all, the user creates a schema defining which entities will be selected from the SBML model (Fig. 2). Building a schema requires certain rules, such as mapping the names of different items found in the schema to their corresponding names as defined in the SBML specifications. To illustrate this principle, node labels and node properties indicated in the schema need to match the name of the SBML component and the properties belonging to the component, respectively. Given this information,
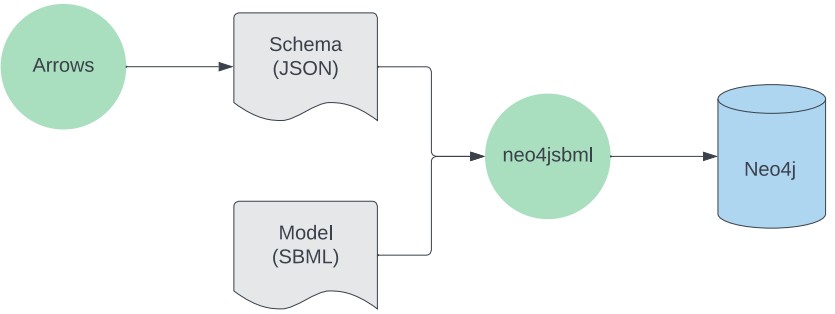

**Figure 1  Workflow for utilizing neo4jsbml.** First, a schema is created with https://arrows.app. Next, aided by the schema, the SBML data is loaded into the Neo4j database using neo4jsbml. Softwares are green, files are gray, the database is blue.

neo4jsbml identifies the objects and attributes to extract from the SBML, thanks to the library libsbml. Thus, only the entities required for analysis were loaded into Neo4j. SBML specifications indicate which components are linked to each other by storing an identifier corresponding to another component as an attribute. Therefore, neo4jsbml checks whether the two components are linked by following the strategies shown in Fig. 3. However, the Neo4j database operates on a directed graph; therefore, neo4jsbml infers directionality as indicated by the schema. Once the data are imported into Neo4j, the entities and relationships from the SBML model can be queried by Cypher through Neo4j.

## Use cases

To illustrate the power of neo4jsbml, three use cases were performed involving three *Escherichia coli* GEMs. The first one, called iAF1260, was published in 2007 (*Feist et al., 2007*). The second one, named iML1515, was established in 2017 (*Monk et al., 2017*). The last one, a small-scale model baptized e_coli_core was derived from an *E. coli* model. All models were downloaded from the BiGG Models knowledgebase (*King et al., 2016*) and the MetaNetX database (*Moretti et al., 2021*).

A Neo4j plugin, named neo4jefmtool, based on efmtool (*Terzer & Stelling, 2008*) was created to enumerate the metabolic pathways, available at https://github.com/brsynth/neo4jsbml.

The schema described in Fig. 2A was used to import data into Neo4j for the first two use cases, whereas the schema corresponding to Fig. 2B was used for the last case. Both schemas were created using Arrows, available at https://arrows.app/. To provide a general overview of the use of neo4jsbml, some examples of models embedded in the SBML specifications are shown in Figs. S1, S2, and S3.

Neo4j version 5.12 and neo4jsbml version 0.12.0 (10.5281/zenodo.8419209) were used for this study.

## RESULTS

We developed neo4jsbml as a modular Python package that can be used as a standalone program or incorporated as a library in other programs. It is available through the Conda
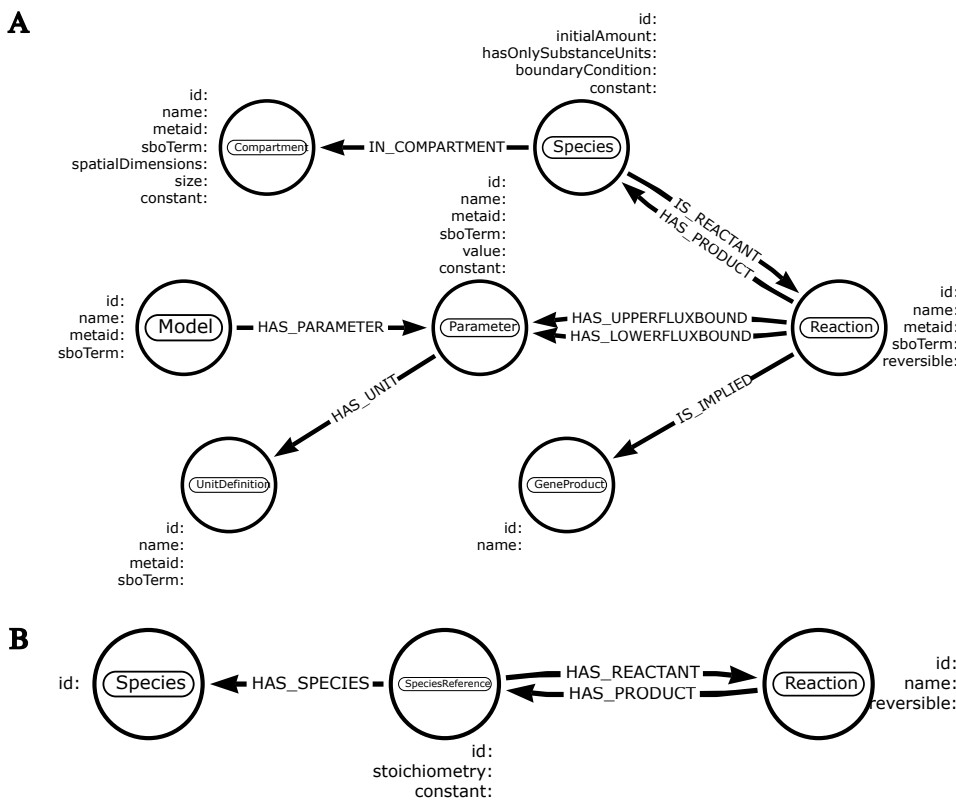

**Figure 2** Schemas created from https://arrows.app. The entities and their relationships are represented by a circle and an arrow, respectively. Node label matches the name of a SBML component. Each node embeds the targeted properties. Types associated to each property are optional. (A) This schema represents various entities found in genome-scale models. (B) This schema is focused on the extraction of a metabolic network.

package management system (*conda-forge community, 2015*) and runs on all platforms without requiring extra privileges. SBML files are loaded by neo4jsbml, using the libsbml library (*Bornstein et al., 2008*), and the selected data will be loaded into the Neo4j database using the Python Neo4j driver, based on a database schema. Twelve packages are part or will be part of the SBML standard, but only eight have been fully implemented by the library libsbml. Neo4jsbml can consider four plugins: Flux Balance Constraints (*Olivier & Bergmann, 2018*), Groups (*Hucka & Smith, 2016*), Layout (*Gauges et al., 2015*) and Qualitative Models (*Chaouiya et al., 2015*) (Table S1). Importing data into Neo4j is done through pure Cypher queries, with no additional Neo4j plugins necessary.

To demonstrate the usefulness of neo4jsbml, we conducted a proof of concept study with three use cases.

## Comparing two models

We compared two versions of the *E. coli* genome-scale metabolic model to highlight improvements between an older and a newer version from two repositories: BiGG and MetaNetX. Firstly, we compared the number of nodes and relationships in both models
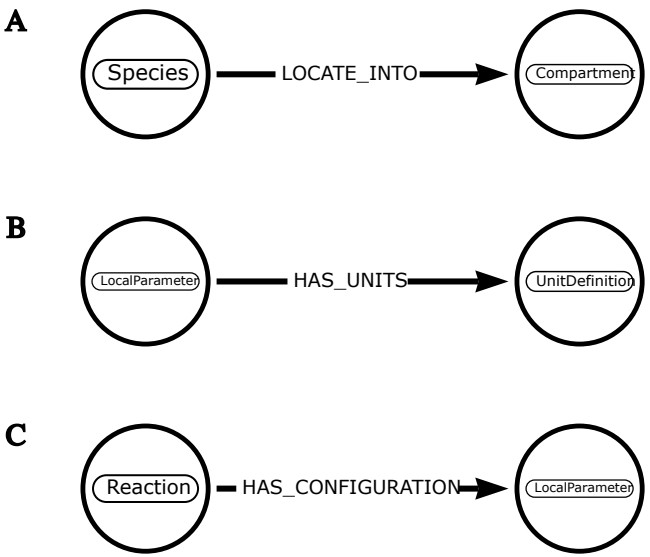

**Figure 3** **Examples of schemas illustrating how neo4jsbml uses introspection to associate one entity with another.** (A) The object *libsbml.Species* has a method named *getCompartment()* that retrieves the identifier of the *libsbml.Compartment* object. The mapping between these entities is established by calling a method based on the name of one entity. (B) The object *libsbml.LocalParameter* has a method named *getUnits()* that retrieves the identifier of the *libsbml.UnitDefinition* object. The linkage between these entities is possible thanks to the name of the relationship. (C) The objects in the libsbml library have a method named *getListOfAllElements()* that leads to list some nested components. The *libsbml.Reaction* object can host several objects, including a *libsbml.LocalParameter* object. The relationship is created according to the name of the entities.

(Table 1). More than four thousand entities and twenty thousand relationships were loaded into Neo4j from the *E. coli* genome-scale models, iML1515 and iAF1260. Given one version of *E. coli*, the number of chemical species and reactions differed between the BiGG and MetaNetX databases. In addition, the iML1515 genome-scale metabolic model contained more nodes and relationships compared to the iAF1260 genome-scale metabolic model.

Next, we analyzed the reactions associated with the fumarate metabolite (Fig. 4). The Cypher query is described in Eq. (1).

$$\text{MATCH } p=((c:Compartment)<-(n:Species \{id: "M\_fum\_c"\})->(r:Reaction)) \qquad (1)$$
$$\text{RETURN } p$$

One additional reaction was linked to the fumarate metabolite in the iML1515 model compared to the iAF1260 model.

## Viewing a metabolic pathway

A common application with a metabolic network is to visualize biological pathways derived from genome-scale models. In this case, the phosphoenolpyruvate metabolite was targeted to identify the chemical species in the extracellular compartment that produce it through

**Table 1** Number of entities and relationships loaded in Neo4j for the iAF1260 and iML1515 GEMs provided by the BiGG and MetaNetX databases. Nodes are in lowercase, relationships are in uppercase.

| Node/Relationship | BiGG | | MetaNetX | |
|---|---|---|---|---|
| | iAF1260 | iML1515 | iAF1260 | iML1515 |
| Compartment | 3 | 3 | 4 | 4 |
| Species | 1668 | 1877 | 1976 | 2217 |
| Parameter | 8 | 5 | 7 | 5 |
| Reaction | 2382 | 2712 | 2374 | 2704 |
| Model | 1 | 1 | 1 | 1 |
| UnitDefinition | 1 | 1 | 1 | 1 |
| GeneProduct | 1261 | 1516 | 1262 | 1517 |
| IN_COMPARTMENT | 1668 | 1877 | 1976 | 2217 |
| HAS_PRODUCT | 4714 | 5328 | 4920 | 5529 |
| HAS_LOWERFLUXBOUND | 2382 | 2712 | 2374 | 2704 |
| HAS_UPPERFLUXBOUND | 2382 | 2712 | 2374 | 2704 |
| HAS_PARAMETER | 8 | 5 | 7 | 5 |
| HAS_UNIT | 8 | 5 | 7 | 5 |
| IS_IMPLIED | 3747 | 4624 | 3750 | 4627 |
| IS_REACTANT | 4517 | 5247 | 4553 | 5242 |

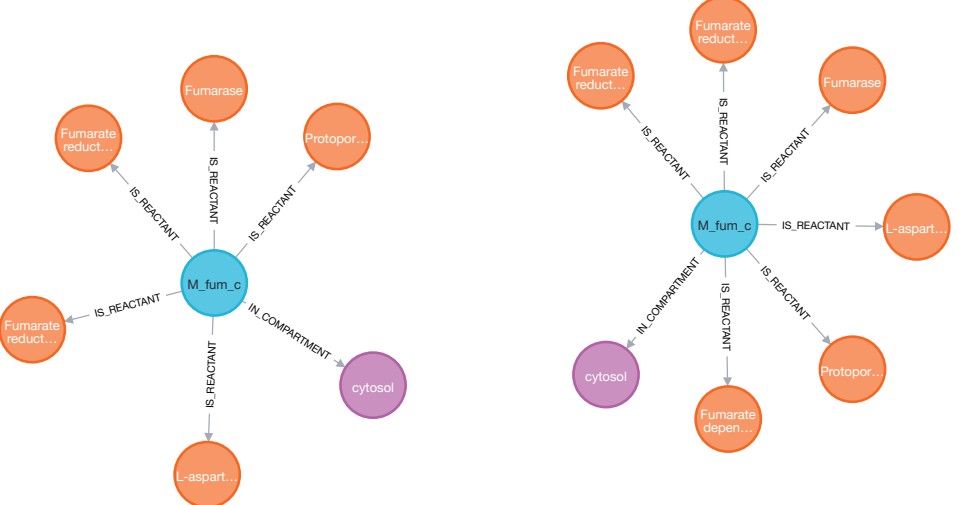

**Figure 4** Visualization of the fumarate metabolite (blue) and its nearest neighbors: reactions (orange) and compartment (purple) in two models (iAF1260 on the left and iML1515 on the right) sequentially imported into Neo4j by neo4jsbml. The reaction, *Fumarate dependent DHORD*, was only shown in the iML1515 model.

exactly two reactions in the core model of *E. coli*. Firstly, the entire genome-scale model was imported into Neo4j.

```
MATCH (s:Species)
WITH s, size ([p=(s)-[:IS_SUBSTRATE]->() | p])
  AS sz ORDER BY sz DESC
WHERE sz > 10
SET s:Hub
```
(2)

```
MATCH path=(:Compartment)<-[:IN_COMPARTMENT]-(s1:Species {id: "M_pep_c"})
  <-[:HAS_PRODUCT|IS_SUBSTRATE*2]-(s2:Species)
  <-[:HAS_PRODUCT|IS_SUBSTRATE*2]-(s3:Species)
  -[:IN_COMPARTMENT]->(:Compartment)
WHERE
  NOT "Hub"
    IN apoc.coll.flatten([n in nodes(path) | labels(n)])
  AND (s3)-[:IN_COMPARTMENT]->(:Compartment {id: "e"})
  AND NOT (s2)-[:IN_COMPARTMENT]->(:Compartment {id: "e"})
RETURN path
```
(3)

Then, the reactions, metabolites, and compartments involved in the metabolic pathway were extracted using two consecutive Cypher queries: dense nodes were flagged Eq. (2) and the paths were selected Eq. (3). With the constraints defined above, the Pyruvate metabolite was identified as a key player (Fig. 5).

## Enumerate metabolic pathways

The enumeration of metabolic pathways identifies and lists all the possible routes of biochemical reactions within a metabolic network. From the core model of *E. coli*, pathways involving the Formate and the Acetaldehyde metabolites were searched. *Species*, *SpeciesReference* and *Reaction* entities were loaded into Neo4j from the genome-scale model.

```
MATCH (n:SpeciesReference)-[:HAS_SPECIES]->(s:Species)
  SET n.id = s.id
MATCH (n:Species) DETACH DELETE n
MATCH (n:SpeciesReference) REMOVE n:SpeciesReference
  SET n:Species
```
(4)

```
MATCH (s:Species)
WITH s, size ([p=(s)-[:HAS_REACTANT]->() | p])
  AS sz ORDER BY sz DESC
WHERE sz > 4
SET s.hub = true
MATCH (s:Species)
WITH s, size ([p=(s)<-[:HAS_PRODUCT]-() | p])
  AS sz ORDER BY sz DESC
WHERE sz > 4
SET s.hub = true
MATCH (s:Species) WHERE s.hub DETACH DELETE s
```
(5)

(6)
```
CALL brsynth.enumeratePathway(["M_for_c", "M_acald_c"], "ep")
```

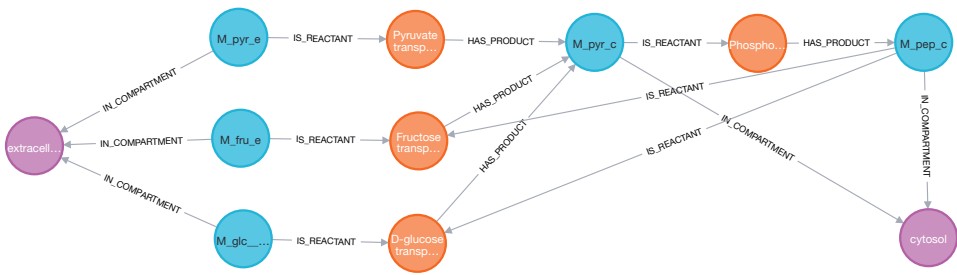

**Figure 5** Thanks to Neo4j and neo4jsbml, the phosphoenolpyruvate metabolite and its precursors implicated in exactly two reactions in the *Escherichia coli* core model (e_coli_core) were visualized. The Pyruvate metabolite was directly involved in the production of Phosphoenolpyruvate without any circular dependency.

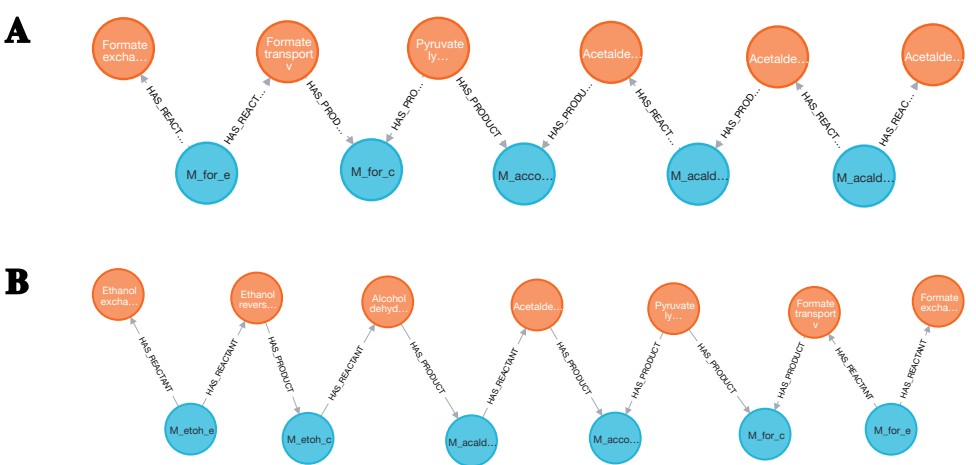

**Figure 6** Two pathways were identified in the *E. coli* core model. (A) A first pathway linking the Formate to the Acetaldehyde metabolites was extracted. (B) A second pathway linking the Formate to the Ethanol metabolites was identified.

Then, graph reconstruction was performed by merging the *Species* and *SpeciesReference* nodes Eq. (4) and removing *Species* that were linked to fewer than four reactions Eq. (5). Next, the neo4jefmtool plugin was used to enumerate the pathways involving the Fumarate and the Acetaldehyde metabolites Eq. (6). The two pathways identified are shown in the Fig. 6.

## DISCUSSIONS

Comparing genome-scale models, viewing metabolic networks and enumerating pathways are common applications in systems biology. On the one hand, biological processes are stored in the SBML format. On the other hand, Neo4j is a powerful graph database suited to link related entities, using the Cypher language. Neo4jsbml aims to conciliate both. The first use case showed the capability to perform quantitative and qualitative analysis on genome-scale models. Neo4j has a comprehensive set of features, as described by

*King et al. (2015)* which include the ability to navigate and search through visualizations, design and customize pathway maps, and represent diverse data types. The second example highlights the ability of Neo4j to serve as an alternative viewer for visualizing metabolic pathways. The last illustration emphasizes the capacity of neo4jsbml to retain the required data and the strength of Neo4j to conduct specialized analyses.

Neo4jsbml employs a novel approach based on the creation of schema entities to identify entities that are useful for a dedicated application. Introspection is a programmatic principle that maps the schema of certain entities to a model to extract desired data. Therefore, neo4jsbml can deal with Level 2 Version 5 of the SBML standard (Fig. S3) and partially with SBML packages (Fig. S2). Our aim was to develop a tool that is as simple as possible, requiring only command-line skills.

However, as we observed for the use cases, the more complex the application, the more intricate the Cypher queries to perform graph refactoring or to extract the data. Additionally, programming skills may be required to develop extensions to perform dedicated analyses. In rare circumstances, data imported into Neo4j can produce unexpected results. The first scenario is shown in Fig. S3J. The model has two *Event* entities, each with one *Event Assignment* entity that supports the *variable* property. Without setting an *id* property for each *Event Assignment*, in the model, it would be difficult to infer the wishes of the user. The second scenario concerns the model presented in Fig. S2C. The *Layout* entity is linked to its *Dimensions* entity as well as to all other *Dimensions* entities found in the model. In these cases, a prior modification of the model or graph refactoring in Neo4j is required to meet expectations. In addition, if several models are loaded sequentially into the database, some entities can be the same, and the relationships between the models are mixed. Neo4jsbml alleviates this difficulty by adding an additional identifier to all imported entities.

The SBML standard represents mathematical expressions using MathML and annotations using the Resource Description Framework by adding XML elements to the model. Neo4jsbml extracts mathematical expressions as strings whereas annotations are loaded with XML tags.

However, neo4jsbml is highly dependent on the web application Arrows to create the schema of entities. To ensure a long-term future, neo4jsbml will need to support at least one other tool to provide the schema of entities. In addition, neo4jsbml uses simple Cypher queries to import data into Neo4j. It would be valuable to load data into another graph database using Cypher.

More broadly, introspection-based programming patterns can be used to extract information from other standards and import the required data into sql, no-sql or graph-oriented databases.

## CONCLUSIONS

Neo4jsbml is a user-friendly Python package and it takes advantage of existing software to define the database schema, parse SBML files, and import the data into the Neo4j database. The three use cases demonstrated the benefits of visualizing SBML data as a graph. The

biological processes, specified thanks to SBML standard, contain intricate interactions between biological entities, and Neo4j's graph represents these relationships with nodes and edges, making it easier to analyze and visualize complex biological networks. Using Cypher, exploring data through statistics or creating subgraphs was intuitive. With the help of the schema database and introspection concept, only relevant information is loaded into the database. We foresee neo4jsbml as an essential tool for bridging the gap between SBML data and Neo4j. It will enable researchers to visualize and analyze relationships between different entities and fully leverage the capabilities of Neo4j.

## ACKNOWLEDGEMENTS

We thank the CATI SysMics workgroup from INRAE. We thank the editor, Ron Henkel, and the anonymous reviewers for their valuable insights.

### Funding

This work was supported by a French government grant managed by the Agence Nationale de la Recherche under the France 2030 program, reference ANR-22-PEBB-0008. The funders had no role in study design, data collection and analysis, decision to publish, or preparation of the manuscript.

### Grant Disclosures

The following grant information was disclosed by the authors:
A French government grant managed by the Agence Nationale de la Recherche under the France 2030 program: ANR-22-PEBB-0008.

### Competing Interests

The authors declare there are no competing interests.

### Author Contributions

- Guillaume Gricourt conceived and designed the experiments, performed the experiments, analyzed the data, prepared figures and/or tables, authored or reviewed drafts of the article, and approved the final draft.
- Thomas Duigou conceived and designed the experiments, performed the experiments, analyzed the data, authored or reviewed drafts of the article, and approved the final draft.
- Sandra Dérozier conceived and designed the experiments, performed the experiments, analyzed the data, authored or reviewed drafts of the article, and approved the final draft.
- Jean-Loup Faulon conceived and designed the experiments, analyzed the data, authored or reviewed drafts of the article, and approved the final draft.

## Data Availability

The software is available at GitHub, Zenodo, and Anaconda:

- https://github.com/brsynth/neo4jsbml

- Gricourt, G. (2023). neo4jsbml dataset [Data set]. Zenodo. https://doi.org/10.5281/zenodo.10245426

- https://anaconda.org/conda-forge/neo4jsbml.

## Supplemental Information

Supplemental information for this article can be found online at http://dx.doi.org/10.7717/peerj.16726#supplemental-information.

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
