# Peer review of "neo4jsbml: import systems biology markup language data into the graph database Neo4j"

_PeerJ, doi:10.7717/peerj.16726_

## Round 0.1 · original submission · Major Revisions

Kindly address all the issues raised by the reviewers, with particular attention to those from reviewers 1 and 3. Failing to do so could result in rejection.

·

Basic reporting

The authors developed "neo4jsbml" to bridge the gap between SBML data and the Neo4j database, allowing users to visualize and analyze complex biological networks with ease. The tool imports only the desired data from SBML files into the Neo4j database using a user-defined schema. The publication demonstrates two use cases, comparing different genome-scale metabolic models and visualizing a metabolic pathway, to showcase the tool's functionality and advantages.

The authors use appropriate scientific terminology and concepts throughout the paper, enhancing the readability and comprehension of the content. The publication provides a coherent explanation of the problem, methodology, and results, making it accessible to the target audience of researchers and scientists in the field of systems biology and bioinformatics.

Experimental design

The authors might benefit from conducting a more comprehensive literature analysis to situate their work in the context of existing research. For instance, the paper "Combining computational models, semantic annotations, and simulation experiments in a graph database" already addresses the transfer of SBML (and other markup languages) to Neo4J using Java. Although the authors' approach covers a broader range by accommodating plugins for SBML L3V2, this aspect remains unexplored in the paper. It may be worthwhile for the authors to elucidate the limitations imposed by libsbml on handling plugins.

Validity of the findings

While the concept of providing users with the ability to define their own schema is intriguing, it is essential to consider its practicality. Few modelers may opt to undertake this task. As a recommendation, pre-developed schemas for SBML L3V2 (and lower versions) utilizing SBML modules/plugins (as mentioned in the conclusions) could be proposed and shiped with this tool to enhance usability.

The authors claim that introspection allows for the loading of only relevant information, yet they need to clarify how this aligns with user-defined schemas or libsbml.

Regarding loading multiple models simultaneously, the high point of their achievement is not explicitly apparent, and further clarification would be beneficial.

Similarly, the authors state that Neo4j is a powerful tool for visualizing SBML data, but it is essential to acknowledge that the built-in viewer might not be suitable for inexperienced users.

Additional comments

Despite these points, I believe that the publication of this paper would be valuable to the community. The flexible Python library presented holds promise and can be a significant asset for researchers. To strengthen their work, the authors should consider the following aspects:

1. Conduct a literature analysis and elucidate how their approach compares to existing research.

2. Clearly highlight the strong points and novelty of their work based on the findings from the literature analysis.

3. Strengthen the argument for the usefulness of their approach and how researchers can benefit from it.

4. Develop additional schemas, particularly covering SBML L3V2 and modules/plugins supported by libsbml.

5. Consider the role of annotations in SBML and explore how to incorporate this valuable resource of information.

6. Showcase the inclusion of other languages (e.g., SBOL, CellML, NeuroML) using their approach, which could significantly enhance the appeal of their work.

Reviewer 2 ·

Basic reporting

1. The Introduction needs several additional references. In particular, it would be useful to reference some works illustrating examples of applications of Genome-Scale Models using SBML.

2. SBML is not only used for representing GEMs. It can be used to model many different types of biological processes and pathways. Again, it would be useful to provide some references of examples where SBML has been used to model some specific biological problems, for instance signaling pathways, cancer-related processes.

3. It would be useful to mention that SBML models are used not mainly as a sort of database, but as models that can be simulated, analysed, optimised, genetically engineered, etc. For instance, examples of flux balance analysis, knock-out analysis, dynamics simulation using tools like COPASI, and so on. I think that the authors should emphasize the flexibility and capabilities of those models to help perform a variety of analyses, as there is a vast amount of literature about this subject.

4. The Supplementary tables could be shown in the main text if this is allowed by the Editorial formatting rules.

Experimental design

5. It needs to be made clear how introspection is used. The authors mentioned that they used some introspection tools from Python, but no details are given. Since this is a technical note, this type of detail is needed.

6. The way Arrows works (shown in Figure 1 and Figure 2) needs to be better explained. It is not very clear the level of automation between the Schema generated by Arrows and the parsing of the SBML model and its conversion to a Neo4j database. How much Python custom code is required? What is the advantage of using Arrows?

Validity of the findings

7. Is it possible to use the resulting Neo4j database to enumerate the pathways between two metabolites?

8. What is the main novelty compared with biochem4j https://doi.org/10.1371/journal.pone.0179130, which already allowed extracting the GEM querying by organism? In particular, biochem4j contained sequence and chemical information, which might be missing in sbml4j. Are there any plans to reconciliate both databases?

9. I can see as a limitation in the current implementation the dependence on the metabolite and reaction labels, which are not always unified. I believe that a more ambitious plan is missing, where sbml4j can be integrated with databases like MetaNetX and therefore, it can go beyond mapping GEMs.

10. Making a stronger case about the usefulness of mapping SBML to Neo4j is needed, especially in the conclusions. In the way the Conclusions are now written, it is not possible to see the advantages of having a GEM represented as a graph database.

Additional comments

11. Have the authors considered the inverse problem, i.e., neo4j -> sbml? I am not sure how easy/challenging will be to implement this feature in the Python package, but it would be very useful. For instance, a researcher can modify the neo4j graph database in order to include some genetic interventions (knock-outs, engineered pathways) and might desire to convert it back to SBML in order to perform standard analysis like FBA. Having the ability to convert it back to SBML will save a lot of work and make the neo4jsbml a more powerful tool.

Reviewer 3 ·

Basic reporting

However, a section is dedicated to "Material and Methods", the research methodologies are not expounded.
The experimental design and development can be categorized by the Frascati manual as software development within R&D. Thereby, the adequate research methodologies are Design Science Research paradigm and Software Case Study methodology.
The research questions and goals according to the IMRAD standard should be formulated.

Experimental design

The genre of the research and the paper can be formulated as software development. It can be considered as research if it resolves a scientific and technological uncertainty, and then a systematic investigation and a comparative study are performed to increase of stock of knowledge.
The paper lacks the explicit formulation of the research questions.
The design of the experiment is missing according to the Design Science Research paradigm.
There was no extensive and intensive testing about the successful path of the solution and cul-de-sacs
The paper is only a plain description of a database and software development project.

Validity of the findings

Since such database development projects can be implemented, the communicated project description is valid for that reason.

Additional comments

The usability and utility of the created database for the end-users should be investigated according to the Software Case Study Methodology.
It would be worth investigating in the literature research whether there is any ontology that could be relevant to database structure and how the database and ontology could be exploited for research in bioinformatics.
Science is about comparison so the results should be compared to previous publications to underpin that this work makes sense.

Reviewer 4 ·

Basic reporting

The paper is easy to understand -- with a clear goal, good introduction, and relevant references. Figures are good quality and well-explained. The authors cite their data sources and software versions

Suggestions for improvement:
Since this manuscript is fundamentally about presenting a new software tool, please clearly include in the paper a source/repository where the code is hosted (https://github.com/brsynth/neo4jsbml). Alternatively, consider adding the command "conda install -c conda-forge neo4jsbml" to line 100: "it is available through the Conda package management system", or to the supplementary section, so a user who is not well-versed with Conda might be easily able to access the tool.

Experimental design

The paper integrates well known software like sbml & neo4j, which adds confidence to its credibility. The methodology for creating and using neo4jsbml follows established software development and database management practices, with well-documented steps for schema creation, data loading, and querying.

Validity of the findings

The validity of the paper is clearly illustrated with the effectiveness and functionality of the neo4jsbml tool, which is communicated through the Use Cases presented by the authors.

Suggestions for improvement:
I encourage the authors to split the Discussion into its own section. Within that section, please discuss the following:
1) limitations or challenges encountered during the development and application of neo4jsbml.
2) emphasize the broader impact of the tool
3) provide more specific ideas for future directions of neo4jsbml

Additional comments

Minor spelling/grammatical errors:
line 41: Replace "embedding" with "embed"?
line 74: Replace "It's" with "It is"
line 99: Replace "developped" with "developed"
line 117: Replace "analyzes" with "analysis"

---

## Round 0.2 · Minor Revisions

Kindly check the reviewers minor comments and address them.

·

Basic reporting

The reviewer expresses gratitude to the authors for their meticulous consideration of the comments and subsequent modifications to the manuscript. All previously highlighted concerns have been diligently addressed, resulting in a notably improved readability of the document. However, there remain some minor observations, enumerated below:

Figure 1: Consider rotating the database symbol by 90°, as its current horizontal orientation appears peculiar.

Figures 2, 3, and Table 1: Ensure uniformity in the style of labels within the nodes, either in lowercase or uppercase. If the labels are derived from the model itself, kindly provide clarification. Additionally, extend this style consistency check to supplementary figures.

Listings: While acknowledging that the font choice may be predetermined by the journal, it is suggested that a typewriter style be considered for code listings. To enhance readability, contemplate incorporating colored coding for key terms such as 'match,' 'where,' etc. Additionally, rectify the orientation of opening quotation marks, which currently face left and appear inconsistent.

Experimental design

All previous concerns have been handeled well.

Validity of the findings

All previous concerns have been handeled well.

Additional comments

The reviewer expresses gratitude to the authors for their dedicated efforts and the substantive revisions made to the document. Acknowledging the likely considerable workload undertaken within a constrained timeframe, it is noteworthy that the manuscript has achieved a commendable level of readability. The reviewer is content with the current state of the manuscript and looks forward to its publication.

---

## Round 0.3 · accepted · Accept

Thanks for addressing those remaining reviewer concerns